# The Effect of Die Material on the Crown Fracture Strength of Zirconia Crowns

**DOI:** 10.3390/ma17051096

**Published:** 2024-02-28

**Authors:** Akram Sayed Ahmed, Nathaniel C. Lawson, Chin-Chuan Fu, Pranit V. Bora, Edwin Kee, Amir H. Nejat

**Affiliations:** 1Department of Dental Biomaterials, Faculty of Dentistry, Tanta University, Tanta 6624033, Egypt; akram_gad@dent.tanta.edu.eg; 2Division of Biomaterials, University of Alabama at Birmingham School of Dentistry, Birmingham, AL 35209, USA; pvbora@uab.edu; 3Division of Prosthodontics, University of Alabama at Birmingham School of Dentistry, Birmingham, AL 35209, USA; ccfu@uab.edu; 4Division of Prosthodontics, LSU School of Dentistry, New Orleans, LA 70119, USA; ekee@lsuhsc.edu (E.K.); anejat@lsuhsc.edu (A.H.N.)

**Keywords:** zirconia, crown fracture, 3D printing

## Abstract

Background: Determination of the eligibility of several tooth analog materials for use in crown fracture testing. Methods: A standardized premolar crown preparation was replicated into three types of resin dies (C&B, low modulus 3D printed resin; OnX, high modulus 3D printed resin composite; and highest modulus milled resin composite). 0.8 mm zirconia crowns were bonded to the dies and the maximum fracture load of the crowns was tested. Twelve extracted human premolars were prepared to a standardized crown preparation, and duplicate dies of the prepared teeth were 3D printed out of C&B. Zirconia crowns were bonded to both the dies and natural teeth, and their fracture load was tested. Results: There was no statistical difference between the fracture load of zirconia crowns bonded to standardized dies of C&B (1084.5 ± 134.2 N), OnX (1112.7 ± 109.8 N) or Lava Ultimate (1137.5 ± 88.7 N) (*p* = 0.580). There was no statistical difference between the fracture load of crowns bonded to dentin dies (1313 ± 240 N) and a 3D-printed resin die (C&B, 1156 ± 163 N) (*p* = 0.618). Conclusions: There was no difference in the static fracture load of zirconia crowns bonded to standardized resin dies with different moduli or between a low modulus resin die and natural dentin die.

## 1. Introduction

Although not common, complete fracture and chipping of both metal ceramic and all ceramic crowns are reported as technical failures for single unit crowns. The five year cumulative fracture rate ranges from 0.05% for metal ceramic crowns to 0.4% for zirconia crowns to 2.3% for lithium disilicate crowns. The five year cumulative chipping rate ranges from 3.1% for zirconia to 2.6% for metal ceramic to 1.5% for lithium disilicate [1]. When new variations of restorative materials or surface treatments are developed, laboratory testing of their effect on the strength must be performed to ensure they do not increase the clinical incidence of crown fracture or chipping.

A common method used to evaluate the strength of dental materials by industry and regulatory agencies is to follow the International Standards Organization (ISO) protocols for assessing physical and mechanical properties [2]. These standards typically represent engineering methods that simplify clinical conditions into basic geometric specimens used to calculate inherent material properties. Eventually, these material properties (i.e., strength and modulus) can be programmed into modeling software in order to understand more complex and clinically relevant geometries [3,4,5].

The ISO standard which describes the protocols for determining the strength of dental ceramic materials (ISO 6872:2015) (International Organization for Standardization, Dentistry-Ceramic Materials) employs three different geometries of specimens: three-point bend flexural, four-point bend flexure, and biaxial flexure (piston-on-three-ball). In the three- and four-point bending test, a bar-shaped specimen is supported on its two ends and loaded at its midpoint (three-point) or at two points centered about its midpoint (four-point). In the biaxial flexural test, a disc-shaped specimen is supported by three balls and is loaded in its center. There are advantages and disadvantages of either test. Four-point bending may be preferred over three-point bending as it places a larger area under the maximum bending moment (the area between the two loading points). This arrangement allows a greater probability that a critical flaw will be present in the maximum bending moment and that the area of the specimen under this load will be more representative of the entire specimen. As a result, four-point flexural testing generally produces lower strength values than three-point flexural testing [6].

The advantage of biaxial flexural strength testing is that the loading of the specimens does not occur on their edges, but rather at the center of the disc-shaped specimen. The importance of this detail is that producing flawless edges on bars for three-point and four-point flexural strength testing is technically difficult and critical flaws are likely to be introduced into these edges when fabricating specimens. The ISO recommends beveling the edges of these bars to overcome this issue [2]. Biaxial flexural strength values are typically higher than three- and four-point bend flexural strength values due to the elimination of the “edge effect” [6]. Although the ISO tests do not represent clinically relevant geometries, the advantage of using them is to produce consistent results between different testing laboratories.

A method to incorporate clinically relevant geometry into strength testing is to utilize a crown fracture test, in which a crown or coping is cemented to a tooth preparation die. The crown may be loaded on its occlusal surface until fracture with either an increasing static load or a repeated fatigue loading protocol. There are several advantages of crown fracture testing over using ISO geometric specimens. First, fabricating crown specimens rather than bar or disc specimens will introduce flaws that are representative of the fabrication process of how a material is used in a clinical application [7,8]. For example, the grinding process used to fabricate one restorative material may have a different effect on its strength than the grinding process used for another restorative material [9]. Although ISO protocols are used to standardize data between laboratories, a previous round-robin study reported that different laboratories measured two-fold differences in the flexural strength of materials depending on how the specimens were polished [10]. Therefore, the most relevant method to fabricate specimens would be to treat them identically to how they are used in a clinical setting.

Second, the presence of a supporting structure may affect the strength of one material more than another. Some materials may bond better to their substructure die, which more efficiently allows stress transfer [11]. Also, the different mismatch in elastic modulus between the crown and the substructure die may allow some materials to fare better than others [12]. Previous studies have reported that rankings of materials with crown fracture load testing do not correlate with flexural strength testing [13].

When performing crown fracture testing, the most relevant die material to use is a natural tooth. The disadvantage of natural teeth, however, is that they cannot be standardized between specimens due to natural variations in geometry and physical properties (due to variations in mineralization, organic composition, and microstructure orientation). Therefore, substitute die materials have been used in previous studies, including resins (filled and unfilled) and metals. Ideally, a die material would have a similar bonding strength and modulus as enamel. Several studies have compared the effect of die material in static and cyclic load testing. Generally, ceramic crowns fractured at a lower load on resin composite dies than on metal dies [12,14,15,16]. A study comparing the fracture strength of zirconia crowns reported that crowns fractured on resin dies produced a similar strength as enamel dies, whereas metal dies produced a significantly higher strength [14]. Another laboratory study reported that zirconia crowns fractured on porous titanium dies produced a similar fracture strength as those fractured on dentin, whereas those fractured on resin composite produced a slightly lower strength [15].

One disadvantage of producing dies from metal or even resin composite is that it requires access to a milling device. Three-dimensional printing allows the fabrication of multiple specimens faster and more economically than milling. Identification of a 3D-printed resin to use as a tooth substitute for crown fracture testing would simplify the laboratory procedure for this test.

The objective of the study was to compare the fracture load of zirconia crowns bonded to dies of different polymer-based die materials and natural teeth of the same geometry. The null hypothesis was that there would be no difference in the fracture force of the crowns bonded to the resin-based die and natural teeth for either material.

## 2. Materials and Methods

A maxillary premolar typodont tooth was used to form a standardized tooth preparation with a height of 4 mm, 10° taper, and modified chamfer finish line, using a coarse diamond tapered rotary cutting bur (6856.31.016 FG, Brasseler, Savannah, GA, USA). An electric handpiece mounted to a surveyor was used to ensure a standard taper. The prepared tooth was mounted onto a cylindrical acrylic base and scanned in a benchtop scanner (3Shape E3, 3Shape Inc., Copenhagen, Denmark) to produce an .stl file (Figure 1a). The .stl was used to 3D print dies (*n* = 12/material) from two resin composite materials (NextDent C&B, NextDent BV, Soesterberg, The Netherlands) and (OnX A1, SprintRay, Los Angeles, CA, USA) in a 3D printer (Pro95, SprintRay). The dies were cleaned in 91% isopropyl alcohol and post-cured, according to the manufacturer recommendations, with Procure 2 (SprintRay). Dies were also milled from blocks of heat-cured resin composite (Lava Ultimate, 3M, St. Paul, MN, USA) using the inLab MC X5 milling machine (Dentsply Sirona, Charlotte, NC, USA) (Table 1).

A crown was designed to fit on the standardized preparation using a uniform 0.8 mm thickness and 0.02 mm cement space with computer-aided design software (Dental Restorative System 2020, 3Shape Inc.) (Figure 1b). The crowns were milled from 3 mol% yttria-stabilized zirconia (Cercon HT, Dentsply Sirona) (Table 1) in the inLab MC X5 milling Machine (Dentsply Sirona). The crowns were then sintered according to the manufacturer’s recommendations. Zirconia crowns were sandblasted with 50 µm alumina particles (Cobra, Renfert, St Charles, IL, USA) for 10 s at 0.2 MPa in a sandblaster (Basic Eco, Renfert, Hilzingen, Germany) and rinsed with water. The crowns (*n* = 12/group) were bonded to the three different types of dies with an MDP-containing, dual-cure, self-adhesive resin cement (Panavia SA Cement Universal, Kuraray America, New York, NY, USA). A 10N occlusal load was applied to the crowns during their setting time (5 min). Specimens were stored in 37 °C distilled water for 24 h.

Specimens were placed into a fixture in a universal testing machine (Instron 5583, Instron Inc., Norwood, MA, USA) which oriented the long axis of the tooth at 30° from the vertical direction of the loading indenter. A stainless steel indenter with an end curvature of 3.5 mm diameter was centered on the central groove of the crown such that it contacted both the buccal and palatal cusps (Figure 2). Compressive loading was applied at 0.5 mm/min crosshead speed until fracture on the buccal cusp. Fracture load was recorded once a 20% drop in load occurred. Crowns were visually observed to ensure that fracture occurred. The highest load before fracture was recorded as the fracture load. Fracture loads were compared using a one-way ANOVA and Tukey post hoc analysis (alpha = 0.05). The sample size of *n* = 12 was selected, as a smaller sample size (*n* = 8) was able to distinguish statistically different groups in a very similar study in our lab [17]. Additionally, a power analysis was conducted to determine the sample size with 80% power and a 0.05 level of significance, assuming a 15% standard deviation and 25% expected decrease in load. It revealed that 12 specimens per group would be needed to detect the postulated effect size [18].

The three-point bend flexural strength bars (2 × 2 × 25 mm) of the 3D-printed materials were produced using the same fabrication process as described in the previous section. Flexural strength bars of the heat-cured resin composite were fabricated by sectioning the composite into 2 × 2 × 17 mm bars with a circular sectioning saw (IsoMet Slow Speed Saw, Buehler, Lake Bluff, IL, USA). The length of the bars was limited by the maximum length of the blocks. Specimens were polished on all sides with 600 grit SiC paper and stored in water for 24 h at 37 °C. The height and width of the specimens were measured with digital calipers and were placed on a testing fixture that contained 2 mm diameter rod supports with 20 mm (3D-printed resins) or 12 mm (heat-cured resin composite) span length. The specimens were loaded in their center at 1 mm/min with a 2 mm diameter steel indenter until fracture.

Following IRB approval, human-extracted premolars with comparable sizes were selected and mounted in acrylic. The teeth (*n* = 12) were prepared with 4 mm, 10° taper, and modified chamfer finish line using a coarse diamond tapered rotary cutting bur. An electric handpiece mounted to a surveyor was used to ensure a standard taper (Figure 3). The prepared teeth were scanned in a benchtop scanner (3Shape E3) to produce an .stl file for each tooth preparation. The .stl was used to 3D print replica dies from a resin composite material (NextDent C&B) in a 3D printer (Pro95, SprintRay). The dies were cleaned in 91% isopropyl alcohol and post-cured according to the manufacturers recommendations with Procure 2 (SprintRay). A crown was designed to fit on each preparation using a uniform 0.8 mm thickness and 0.02 mm cement space. The crowns were milled from 3 mol% yttria-stabilized zirconia (Cercon HT, Dentsply Sirona) in the inLab MC X5 milling Machine (Dentsply Sirona). The crowns were then sintered according to the manufacturer’s recommendations. Zirconia crowns were sandblasted with 50 µm alumina particles (Cobra, Renfert, St Charles, IL, USA) for 10 s at 0.2 MPa in a sandblaster (Basic Eco, Renfert) and rinsed with water. The crowns (*n* = 12/group) were bonded to the two different types of dies with an MDP-containing, dual-cure, self-adhesive resin cement (Panavia SA Cement Universal, Kuraray America, New York, NY, USA). A 10N occlusal load was applied to the crowns during their setting time (5 min). Specimens were stored in 37 °C distilled water for 24 h. Specimens were placed into a fixture in a universal testing machine and tested as described in the previous section. The fracture force required to fracture the crowns on the tooth dies and resin dies was compared with a *t*-test (alpha = 0.05).

## 3. Results

The fracture force of the zirconia crowns on the three resin die materials are presented in Table 2. Additionally, the flexural strength and elastic modulus of the three resin die materials are presented in Table 2. Normality assumption of the data was verified using a histogram and a Normal Q-Q plot. A one-way ANOVA revealed no significant differences between the fracture force of the zirconia crowns on the three die materials (*p* = 0.580). The crown fracture force of zirconia crowns on natural tooth dies and a 3D-printed resin replica die (C&B) are presented in Table 3 and a *t*-test revealed that there was no significant difference between the two groups (*p* = 0.618).

Examination of the fractured specimens revealed that crowns fractured either through the occlusal groove or close to it (Figure 4a and Figure 5a). There were specimens from each group in which the die fractured along with the crown (Figure 4b and Figure 5b) and this fracture was observed at the interface between the prepared tooth and the base. The fraction of each failure type is presented in Table 4.

## 4. Discussion

The first hypothesis stating that zirconia crowns bonded to stiffer resin-based die materials would achieve higher fracture loads was not accepted. The elastic modulus of the stiffer resins was 2.5–4 × more than the least stiff resin. Despite this difference, there was no statistical difference in the fracture load of the zirconia crowns on any of the resin dies.

The results of the study are similar to a previous study. A study by Machry et al. [12] bonded 0.7 mm and 1.0 mm 3 mol% yttria-stabilized zirconia discs to 2 mm discs of epoxy resin (modulus = 14.9 GPa) and composite resin (11 GPa). The discs were cyclically step-loaded against a stainless steel, spherical piston. The zirconia specimens bonded to the different substrate discs demonstrated a similar fatigue fracture load (maximum fatigue load to cause fracture).

In the study by Machry et al. [12], flat specimens of zirconia were loaded perpendicular to their surface by a round ball. In this configuration, the mismatch in elastic modulus between the zirconia and substructure resins indicates that the substructure may undergo more strain than the zirconia under a given load [8]. As a result, tensile forces accumulate in the undersurface of the zirconia opposite the loading ball. In the configuration used in the current study, a spherical indenter loaded a single cusp at 30° off the axis of the tooth. Observation of the fractured specimens reveals that fractured pieces were often split between the occlusal groove. This observation suggests that load application was transferred to the occlusal groove. A previous finite element analysis of crowns indented by a sphere reported that loading on steeper cusps transfers stress to the occlusal groove, whereas loading a flatter cusp concentrates stress below the indenter [19].

The second null hypothesis stating that zirconia crowns bonded to a resin-based die material would achieve a similar fracture load as those bonded to natural dentin was accepted. The least stiff resin (elastic modulus = 1.9 GPa) was chosen to be compared to dentin (elastic modulus = 19.3 GPa) [5]. The zirconia crowns fractured on the resin dies produced a statistically similar fracture load as those on natural dentin dies.

A previous study by Yucel et al. [14] produced conical dies of epoxy resin (elastic modulus = 11.8 GPa) and dentin (elastic modulus = 18.6). Uniform 0.6 mm thickness zirconia crowns were bonded to the dies. A vertical load was applied with a metal 2 mm diameter indenter until failure. There was no statistically different fracture strength of the zirconia crowns on either the epoxy resin or dentin dies. A study by Jian et al. [15] examined zirconia crowns bonded to dentin, porous titanium (elastic modulus = 18.5 GPa), and resin composite (elastic modulus = 17.3). Crowns were loaded to failure by a 6mm diameter steel ball. Despite the similar elastic modulus of porous titanium and resin composite, the zirconia crowns fractured on porous titanium were significantly stronger. The crowns fractured at a similar load using both porous titanium and dentin dies. A possible explanation is that the porous titanium and dentin dies were stronger and did not fracture during their use, whereas the resin composite dies fractured during testing.

The elastic modulus of zirconia has been reported to be around 200 GPa. The difference in the elastic modulus between zirconia and the die materials is much greater than the difference between the different die materials. This relative difference may account for the similar fracture load of zirconia crowns on the different die materials. A previous study examined the fracture of a resin-based crown material (polymer-infiltrated ceramic network material, Enamic) (Vita Enamic, Vita Zahnfabrik, Germany) on dies with different elastic moduli [20]. The elastic modulus of the resin-based crown material (37 GPa) was closer to that of the die materials. In that study, crowns fractured at the highest load on dentin (elastic modulus = 18 GPa), followed by a dentin analog (G10, elastic modulus = 15 GPa) and then a resin composite (elastic modulus = 10 GPa). Previous studies have examined the crown fracture resistance of lithium disilicate (elastic modulus = 15 GPa) which has a lower modulus than zirconia but a much higher modulus than resin composite [21]. In a fatigue study, lithium disilicate achieved a similar fatigue failure load and the number of cycles until failure on two different die materials (G10, elastic modulus = 20.4 GPa and fiber-reinforced polyamide-nylon, elastic modulus = 18.7 GPa) [19]. In a static fracture test, lithium disilicate crowns fractured at a similar load on dies fabricated from composite resin as dies fabricated from lithium disilicate. The authors noted that crowns fractured on the lower modulus composite dies exhibited a higher coefficient of variation in fracture data and larger fractures than those fractured on lithium disilicate dies [21]. These results suggest that, although low modulus resin materials may be suitable die materials for zirconia, they may not be well suited for lower modulus restorative materials, such as resin-based crown materials. The suitability of resin-based dies for glass-based restorative materials, such as lithium disilicate, is not well determined.

The fracture mode of the specimens with the die (Figure 4b) was unexpected. The fracture of the die could suggest that the die materials do not have adequate strength for this testing protocol and metal dies should be used instead, as observed in the study by Jian et al. [15]. Fracture of the tooth specimens, however, implies that this failure mode has some clinical relevance. Perhaps the reason that tooth/die fracture was observed more commonly in this study is due to the use of premolar teeth which are smaller and weaker teeth.

Aside from the research implication of the study, the clinical relevance of the study is that the modulus of different resin core materials under zirconia crowns may not affect their fracture load. Therefore, a core buildup performed from a low modulus resin may be acceptable under a zirconia crown. Previous studies have shown that metal-based dies allow a higher fracture strength of zirconia crowns [14], suggesting that crowns cemented on cast cores or titanium abutments would be expected to withstand a higher load prior to failure.

The methodology used in this study has several major limitations and therefore this study serves only as a piece of the puzzle to better elucidate a test methodology for crown fracture testing. A blatant disadvantage of this test is that static loading was performed rather than cyclic loading. High static loads applied by blunt spherical indenters produce Hertzian cone cracks located at the contact surface just outside the contact area. When lower stresses are applied cyclically, a single crack may be initiated on the cement surface which is driven towards the contact surface until fracture occurs. The latter form of fracture is more representative of clinical failures [8]. Practically, the importance of fatigue loading a specimen prior to static fracture testing would be to determine if certain groups would be more affected by pre-cracks created by fatigue loading than other groups. A previous review reported that certain ceramic materials are more susceptible to reduced fracture strength than others; however, the review only examined bilayered ceramic crowns [22].

An additional disadvantage of this test is that all testing was performed dry [14]. Water present at crack tips may potentiate fracture. This moisture may be present from saliva on the external surface of the crown or from pulpal fluid on the cement surface of the crown. Therefore, the modeling of crown fracture testing should be performed in a water chamber and should possibly include a wet, porous die material [8]. Due to the complex geometry of the specimens, it is not possible to calculate a stress (strength) from the crown fracture test. But since all the surface areas of the specimens in the groups we are comparing are the same, the differences in load values should be proportional to the differences in calculated strength.

Future testing should examine the use of different dies when tested under wet cyclical loading. Additionally, other ceramic and resin crown materials should be tested, as the results of this study will not translate to other crown materials.

## 5. Conclusions

Within the limitation of the current study, it was concluded that there was no difference in the static fracture load of zirconia crowns bonded to 3D-printed or milled resin dies with moduli ranging from 1.9 to 7.9 GPa, and that there was no difference in the static fracture load of zirconia crowns bonded to a 3D-printed resin die (elastic modulus = 1.9 GPa) or to natural dentin.

## Figures and Tables

**Figure 1 materials-17-01096-f001:**
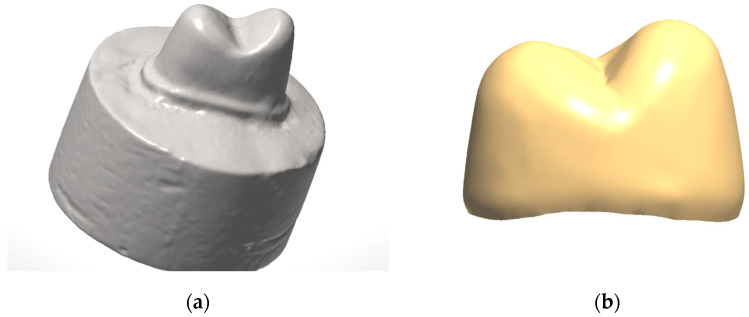
(**a**) Standardized tooth preparation; (**b**) 0.8 mm zirconia crown design.

**Figure 2 materials-17-01096-f002:**
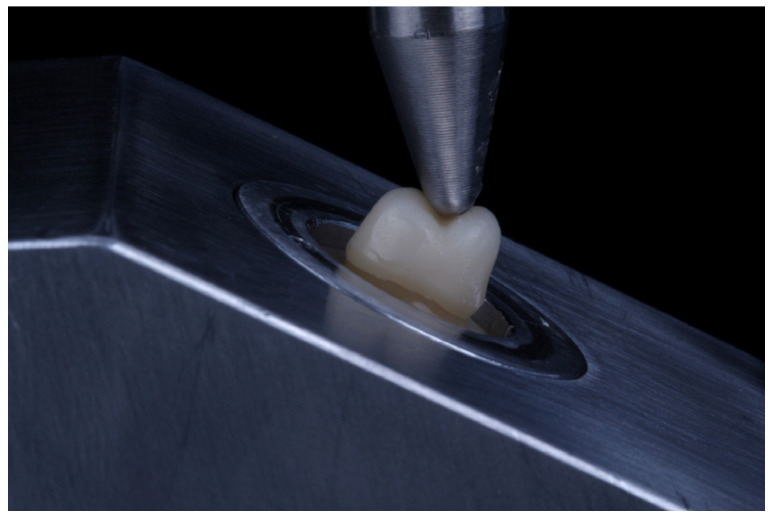
Loading of crowns at 30° off axis by 3.5 mm diameter steel indenter.

**Figure 3 materials-17-01096-f003:**
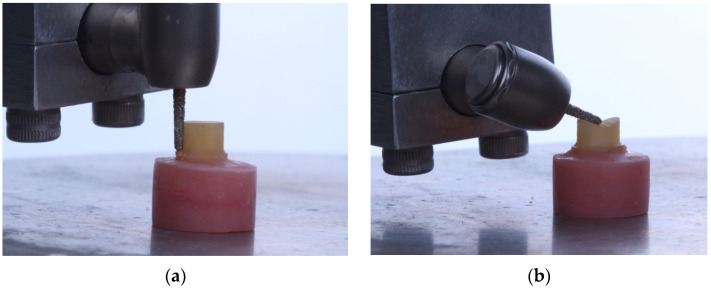
Preparation of natural tooth preparation on its (**a**) axial wall and (**b**) occlusal surface.

**Figure 4 materials-17-01096-f004:**
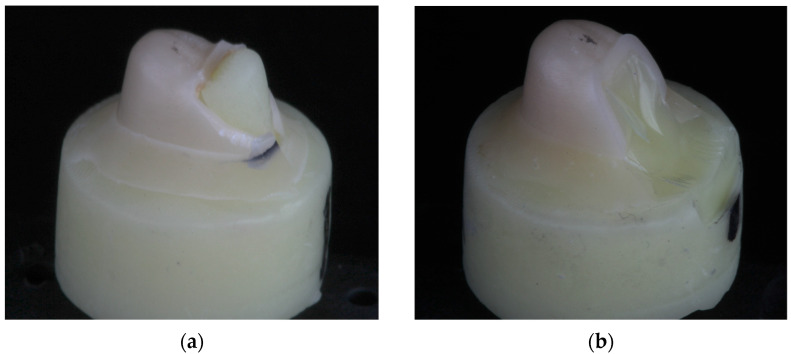
Representative fractured crown (**a**) in which the resin composite die did not fracture and (**b**) one in which it did.

**Figure 5 materials-17-01096-f005:**
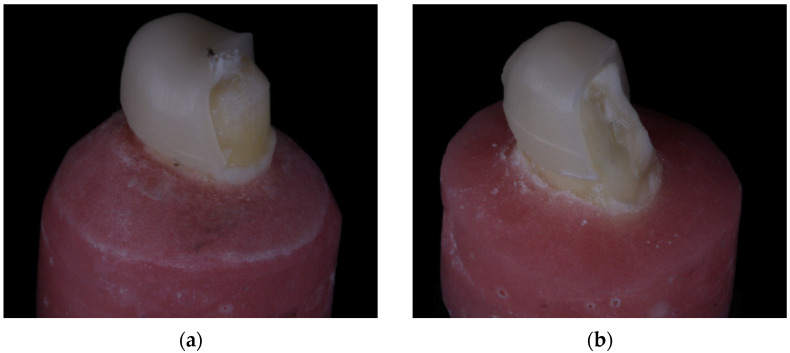
Representative fractured crown (**a**) in which the dentin die did not fracture and (**b**) one in which it did.

**Table 1 materials-17-01096-t001:** Materials used in this study.

Material	Manufacturer	Composition
NextDent C&B	NextDent BV	3D-printed resin
OnX	SprintRay	3D-printed resin
Lava Ultimate	3M	Milled resin
Cercon HT	Dentsply Sirona	3 mol% yttria-stabilized zirconia

**Table 2 materials-17-01096-t002:** Mean ± standard deviation of zirconia crown fracture force on standardized dies, die material strength, and die material elastic modulus.

	NextDent C&B	OnX	Lava Ultimate
Crown fracture force (N)	1084.5 ± 134.2	1112.7 ± 109.8	1137.5 ± 88.7
Three-point flexural strength (MPa)	90.2 ± 8.8 a	116.0 ± 14.7 b	134.4 ± 15.2 c
Elastic modulus (MPa)	1951.0 ± 129.2 a	5063.0 ± 591.4 b	7905.1 ± 1152.5 c

Materials in each row with different letters are statistically significantly different.

**Table 3 materials-17-01096-t003:** Mean ± standard deviation of zirconia crown fracture force on dentin dies and 3D-printed die replica.

	Dentin	NextDent C&B
Crown fracture force (N)	1313.1 ± 240.2	1156.7 ± 163.6

**Table 4 materials-17-01096-t004:** Fraction of types of failure modes of zirconia crowns.

	Fracture through Crown Only	Fracture through Crown and Die
Standardized C&B	8/12	4/12
Standardized OnX	8/12	4/12
Standardized Lava Ultimate	10/12	2/12
Dentin	8/12	4/12
C&B replicas	9/12	3/12

## Data Availability

All source data may be obtained from the corresponding author.

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
