# Peer review of "The Effect of Die Material on the Crown Fracture Strength of Zirconia Crowns"

_materials, 2024, doi:10.3390/ma17051096_

Round 1

Reviewer 1 Report

Comments and Suggestions for Authors

This paper examined whether there would be differences in the fracture force of zirconia crowns bonded to different resin-based dies compared to natural teeth. The topic is interesting, the study was well contacted (sample size calculation, proper samples dimensions, data are well presented etc.) and the limitations are well defined. However, the English language needs very thorough editing and some typos need revision also.

For example, in figure 4 it says “facture” instead of fracture, or in page 7 the phrase “would be expected withstand a higher load prior to failure” should be “would be expected to withstand a higher load prior to failure”. Similarly in the second sentence in introduction section the phrase “are reported technical failures” should be “are reported as technical failures”

In addition, it would be of interest to the readers to see pictures of fractured crowns on the different die materials, including natural teeth.

Comments on the Quality of English Language

It needs moderate editing

Author Response

This paper examined whether there would be differences in the fracture force of zirconia crowns bonded to different resin-based dies compared to natural teeth. The topic is interesting, the study was well contacted (sample size calculation, proper samples dimensions, data are well presented etc.) and the limitations are well defined. However, the English language needs very thorough editing and some typos need revision also.

For example, in figure 4 it says “facture” instead of fracture, or in page 7 the phrase “would be expected withstand a higher load prior to failure” should be “would be expected to withstand a higher load prior to failure”. Similarly in the second sentence in introduction section the phrase “are reported technical failures” should be “are reported as technical failures”

 Thank you.  All grammatical errors have been corrected.

In addition, it would be of interest to the readers to see pictures of fractured crowns on the different die materials, including natural teeth.

Thank you.  We have added a picture of a fractured crown on a dentin die.

Reviewer 2 Report

Comments and Suggestions for Authors

I think the paper is fine.  

Author Response

Thank you.

Reviewer 3 Report

Comments and Suggestions for Authors

The manuscript reports experimental measurements of the fracture load of Zirconia crowns bonded to three standardized dye materials, including C&B, low modulus 3D printed resin; OnX, high modulus 3D printed resin composite; highest modulus milled resin composite. The fracture load was also reported for Zirconia crowns bonded to natural human teeth. The study provided very little data with 1 table (Table 2) and 1 figure (Figure 4) and thus very limited information to the audience. Consequently, the manuscript needs major revision before being accepted for publication in Materials.

Below are the specific comments from this reviewer:

1.      Table 1: The table listed only the materials used in this study. In the results section, however, it was stated that “The fracture force of the zirconia crowns on the three resin die materials are presented in Table 1.”

2.      Table 2: What p value was used to determine the statistical significance?

3.      Figure 4: Please add a table to summarize the detailed fracture modes for each of the 3 dye materials.

4.      Abstract and Conclusion sections: Fracture loads were reported as the main fracture test results. From biaxial flexural test, flexural strength and modulus are much more meaningful.

5.      The difference between in the crown fracture force of zirconia crowns on natural tooth dies and a 3D printed resin die was only briefly described in the results section. Please provide a table to summarize the data details.

Author Response

Below are the specific comments from this reviewer:

  1. Table 1: The table listed only the materials used in this study. In the results section, however, it was stated that “The fracture force of the zirconia crowns on the three resin die materials are presented in Table 1.”

This was a typo. The fracture force of the zirconia crowns on different standardized dies is presented in Table 2 not Table 1.  Thank you for noting this error. It has been corrected.

  1. Table 2: What p value was used to determine the statistical significance?

Alpha = 0.05. Thank you.  This has been added.

  1. Figure 4: Please add a table to summarize the detailed fracture modes for each of the 3 dye materials.

This Table has been added (Table 4).

  1. Abstract and Conclusion sections: Fracture loads were reported as the main fracture test results. From biaxial flexural test, flexural strength and modulus are much more meaningful.

Due to the complex geometry of the specimens, it is not possible to calculate a stress (strength) from the crown fracture test.  But since all the surface areas of the specimens in the groups we are comparing are the same, the differences load values should be proportional to the differences in calculated strength.  This information has been added as a limitation of the study.

  1. The difference between in the crown fracture force of zirconia crowns on natural tooth dies and a 3D printed resin die was only briefly described in the results section. Please provide a table to summarize the data details.

This table has been added (Table 3).

Round 2

Reviewer 3 Report

Comments and Suggestions for Authors

The authors have addressed the all the 1st round review comments.

Comments on the Quality of English Language

The English writing is ok.